# ‘If in Doubt, Sit Them Out’: How Long to Return to Elite Cycling Competition following a Sports-Related Concussion (SRC)?

**DOI:** 10.3390/ijerph20085449

**Published:** 2023-04-10

**Authors:** Neil Heron, Nigel Jones, Christopher Cardwell, Clint Gomes

**Affiliations:** 1Centre for Public Health, Queen’s University Belfast, Belfast BT12 6BA, UK; 2Medical Department, British Cycling, Manchester M11 4DQ, UK; 3School of Medicine, Keele University, Staffordshire ST5 5BG, UK

**Keywords:** concussion, sports-related concussion (SRC), recovery, time, duration, return to sport, return to competition

## Abstract

Introduction: A concussion or sports-related concussion (SRC) is a traumatic brain injury induced by biomechanical forces. After a SRC diagnosis is made, a concussed individual must undergo a period away from competition while they return to their baseline level of functioning. The Union Cycliste Internationale (UCI) currently recommend a minimum of 6 days restriction from competitive cycling following a SRC but there is a growing feeling amongst those involved in brain injury research that this period is too short. Therefore, how much time should cyclists be removed from competitive sporting action following a SRC? Aims: To review the time out of competition following the diagnosis of a SRC for elite cyclists within British Cycling (BC). Methods: All medical records for elite cyclists within BC were audited for diagnoses of “concussion” or “sports-related concussions” from January 2017 until September 2022. The days out of competition following the concussion until ready to compete again (that is, returned to full training) was then calculated. All diagnoses and management of SRC were undertaken by the medical team at BC and in-keeping with current international guidelines. Results: Between January 2017 and September 2022, there were 88 concussions diagnosed, 54 being males and 8 in para-athletes. The median duration for time out of competition for all concussions was 16 days. There was no statistical difference between males (median 15.5 days) and females (median 17.5 days) for time out of competition (*p*-value 0.25). The median duration out of competition following a concussion for able-bodied athletes was 16 (80 athletes) compared to 51 days (8 athletes) in para-cyclists, which was not statistically different (*p*-value 0.39). Conclusions: This is the first study to report SRC concussion recovery times in elite cycling, including para-athletes. Between January 2017 and September 2022, there were 88 concussions diagnosed at BC and the median duration for time out of competition for all concussions was 16 days. There was no statistically significant difference in recovery times between male and females and para- and able-bodied athletes. This data should be used to help establish minimum withdrawal times post-SRC for elite cycling participation and we call on the UCI to review this data when establishing SRC protocols for cycling, with further research required in para-cyclists.

## 1. Introduction

### 1.1. What Is Concussion?

A concussion or sports-related concussion (SRC), referred to as SRC throughout this article, “is a traumatic brain injury induced by biomechanical forces” [1]. The term SRC is sometimes used interchangeably with mild traumatic brain injury, mTBI, although SRC is actually a subset of mTBI [2]. SRCs are common, with one study in Ontario, Canada reporting an average annual incidence of 1153 per 100,000 residents [3]. Within road cycling, head injuries have been found to be the 3rd commonest group of injuries, accounting for between 5% and 15% of all injuries in the sport [4]. Concussions can occur without an obvious head injury and can present with a wide range of signs and symptoms. They are therefore difficult to diagnose and there is no exclusive test currently available to confirm the diagnosis [5].

### 1.2. Symptoms and Risks of Concussion

Concussions carry potential acute and long-term consequences. Acute symptoms of concussion can be split into somatic, cognitive, mood and sleep effects [6]. Acute somatic symptoms include headache, photo-/phonophobia, dizziness, nausea and vomiting, whilst acute cognitive effects can include difficulty with concentration and issues with memory [6]. Acute mood effects include sadness, irritability and nervousness, whilst the impacts on sleep include sleeping too much or too little and difficulty falling asleep [6]. If another brain injury occurs before recovery from the first concussion, then recovery is likely to be prolonged [6]. There is also the small but real risk of second impact syndrome, which carries a high risk of mortality [7]. Thus, it is important for the public and health professionals involved in sport to recognise potential SRCs and remove the player/athlete immediately from the field of play so that a more thorough medical assessment can take place (‘if in doubt, sit them out’).

Some individuals can experience long-term effects following SRC and these include physical and cognitive deficits as well as mood disturbances [8]. The more concussions suffered by an individual, the higher the risk of developing these persistent symptoms [9]. Repetitive concussive events have been associated with the later development of neurodegenerative conditions such as chronic traumatic encephalopathy (CTE) [10]. CTE symptoms include motor dysfunction, cognitive impairment and/or mood disturbances and can present many years after the reported concussion(s) [10]. More recently, research has found that repetitive sub-concussive head impacts may also be associated with an increased risk of neurodegenerative disease [11].

### 1.3. How Are Concussions Diagnosed?

The Sports Concussion Assessment Tool 5 (SCAT5) can be used by health care professionals in the acute evaluation of SRC in adults [12], with the child SCAT5 being used for those between the ages of 5 and 12 years old [13]. The SCAT5 can be augmented by video analysis of the SRC incident as well as other screening modalities [14]. Some sports have evolved this into a head injury assessment (HIA) tool [15]. Newer diagnostic markers, including salvia [16] and blood [17], are the subject of ongoing research but are not yet practicable in the immediate assessment of an individual with suspected SRC. After a diagnosis of SRC, the individual must undergo a period away from competition whilst they return to their baseline level of functioning and become free of symptoms. A graduated return to play (GRTP) protocol must then be successfully completed before a return to competition is permitted. The GRTP process involves the individual being medically guided through a stepwise progression in activities, starting with gentle exercise such as riding on a static bike, and ending with ‘normal’ unrestricted training. The minimum length of time an individual should be kept away from competition following a concussion is a source of debate, particularly when financial gains associated with competing may motivate some to return before their symptoms have fully resolved. Indeed, psychosocial pressures faced by athletes are increasingly being recognised in affecting athletes’ decisions to return to play [18]. Another important consideration when determining an appropriate minimum stand-down period is the likelihood that apparent clinical recovery occurs prior to physiological recovery [19].

### 1.4. Recommended Periods of Time out of Competition following a Concussion

The 1st international conference on concussion on sport used expert consensus to propose a GRTP protocol that permitted a return to competition as soon as 6 days following injury for some adult individuals. It was, and still is, recognised that age affects return to play time, with younger, high school athletes taking longer to recover than older, professional athletes [20]. Return to play times differ between sports although the evidence for this variation is not clear [21,22] and may be influenced by the timing of the next competitive event or game following a concussion. Elite cycling does not have the same pressure to have athletes ready for the following weekend because of the timings of the international competitive cycling calendar, which often has several weeks between important competitions.

The English Football Association state that the earliest a player aged 20 years or older can return to play is 6 days in an ‘enhanced care setting’ [23] and this approach is supported by the world governing body of football, FIFA [24]. It is also echoed by the governing body for cycling, the Union Cycliste International (UCI) [25]. However, in professional rugby, the period out of competition is now 12 days for players not meeting set criteria [26]. We wanted to address the knowledge gap on concussion recovery within elite cycling by reviewing the time period out of competition (that is, the time to return to full, unrestricted training) following a diagnosis of SRC for all elite riders within the Great British Cycling Team (GBCT). This is the first study to report recovery times post-SRC for elite cyclists.

## 2. Aim

To review the time out of competition following the diagnosis of a concussion for all elite cyclists within the Great British Cycling Team.

## 3. Methods

Since 2017, all medical records within British Cycling have been kept on an electronic system (Performance Date Management System—PDMS). This allows the medical team to record diagnoses and then input appropriate medical notes. For this project, we audited all episodes of “concussion” (and “sports-related concussion”) recorded in the medical records for our elite cyclists from January 2017 until September 2022. Training and competition-related injuries were included for analysis but we did not differentiate between the two. All elite cyclists within the GBCT are full-time athletes, representing their country at the highest level of cycling sport, including international events such as the Olympic and Paralympic Games. There are approximately 120 riders each year included within the GBCT pool of riders. For each recorded episode of concussion, we calculated the number of days taken (from the day of injury) for the athlete to be medically cleared for a return to competition. The medical record for each concussion episode was created and updated by two medical doctors caring for the GBCT athletes, NJ and CG, with the SRCs diagnosed and managed as per international guidelines [12,25]. Data were anonymously plotted in Excel sheets and then analysed in the statistical package, SPSS (IBM SPSS 29). An independent statistician, CC, with input from the lead author, NH, analysed the data. The data were assessed for normality by plotting the results on a histogram. The data were described using medians and minimum/maximum values. Between-group differences were tested for using the non-parametric Mann–Whitney U test, with a statistically significant result being *p* < 0.05.

## 4. Results

Between January 2017 and September 2022, there were 88 concussions diagnosed and recorded (Figure 1). The age range of the included cyclists was 18–44 years old, with the majority being from road and track cycling. A total of 8 were para-athletes (7 males and 1 female) and 80 (47 males, 33 females) were ‘able-bodied’ cyclists. The median duration for time out of competition for all concussions was 16 days, with a minimum of 4 days (year 2017) and maximum of 291 (year 2019) (see Figure 1). When comparing the time out of competition between males (median 15.5 days) and females (median 17.5 days) using a 2-sample Mann–Whitney test, there was no statistical difference (*p*-value 0.25). Meanwhile, the median duration out of competition following a concussion for able-bodied athletes was 16 days (80 athletes) compared to 51 days (8 athletes) in para-cyclists. However, there was no statistical difference between these two groups when compared using a 2-sample Mann–Whitney test, with the p-value being 0.39. A total of 18 riders suffered more than one concussion over the study period, with 1 rider having suffered 4 concussion episodes, 7 riders having 3 episodes and 10 having 2 episodes. Due to the small number of athletes involved, no further analysis was undertaken on riders with multiple concussions.

## 5. Discussion

### 5.1. Summary of Findings

This is the first study to report SRC concussion recovery times in elite cycling, including para-athletes. Between January 2017 and September 2022, there were 88 concussions diagnosed in elite GBCT cyclists and the median duration for time out of competition for all concussions was 16 days. There was no statistically significant difference between males (median 15.5 days) and females (median 17.5 days) for time out of competition following a concussion. Additionally, there was no statistically significant difference between able-bodied athletes (median days 16 in 80 athletes) and para-cyclists (median 51 days in 8 athletes) for time out of competition following a concussion, albeit the numbers were low.

### 5.2. Median Time out of Competition Post-Concussion

Wasserman et al. [27] undertook the National Collegiate Athletic Association Injury Surveillance Program describing the epidemiology of SRC across 25 collegiate sports. They describe 1670 SRCs between 2009 and 2014 and most (60.1%) resolved within one week, although there was a trend for increasing time to return to sport following a SRC over the study period. That is, recovery periods post-SRC becoming more conservative with time. Furthermore, Iverson and Gardner [28] reported concussion incidence and recovery times over two seasons in the National Rugby League (NRL) in Australia. They report that the median time for medical clearance to return to competition following a SRC was 6 days, with most players not missing a game following a SRC. However, one difference between competitive cycling and other sports, such as rugby, is that competitions do not generally take place every week, and this may reduce the risk of athletes under-reporting symptoms to be available to compete sooner. There is also a trend of increasing the mandatory time out of competition following a concussion, with World Rugby increasing this period to 12 days for the 2022/23 season unless specific criteria are met [26]. Thus, from reviewing our median time to return to competition of 16 days following a concussion, the UCI may need to reconsider their concussion policy [25] and increase the minimum time out of competition following a SRC beyond the current 6 days.

### 5.3. Females Versus Males for Time out of Competition following a Concussion

Do females have a prolonged recovery compared to males following a concussion? The evidence for sex affecting concussion recovery times appears mixed. In the study by Iverson et al. [29], they undertook an injury surveillance study of 808 collegiate athletes, of mean age 20.4 years old, between 2014 and 2019 in the United States of America. Although women took slightly longer to return to education/school (median of 5 days versus 4 days, *p* = 0.001) there was no difference in return to sport time between males and females. Indeed the retrospective cohort study by Chand et al. [30] followed 431 high school and collegiate athletes, aged between 10 and 21 years old and who had all experienced at least one previous concussion, for their SRC recovery. They report that male and female SRC recovery times do not statistically differ although females appeared more likely to require neuropsychological referral for their SRC management. However, the study by Zuckerman et al. [31] states that females report more symptoms at baseline and take longer to recover to their baseline symptom level than males following a SRC. Our study indicates that females do not take longer to recover from SRC than males in elite cycling. Therefore, is sex a real risk factor for slower recovery following SRC or is it that females are better at communicating their symptoms, potentially leading to a more conservative approach in managing their SRC and returning to sport? Indeed, male rugby league players have been found to under-report concussive events and symptoms to allow them to continue to play and compete [32]. Thus, the influence of sex on concussion duration is not currently clear and requires further research.

### 5.4. Para-Athletes Compared to Non-Disabled Athletes

The first position statement of the Concussion in Para Sport (CIPS) group was published in 2021 [33]. This consensus paper highlighted the lack of data around return to sport decisions in para-athletes and our paper helps to partially address this knowledge gap by describing the median time out of competition following a concussion as 51 days in this group of elite cycling athletes. However, there were only eight athletes in our cohort and there was no statistical difference between their recovery time and that of the able-bodied group of athletes. Further studies involving larger cohorts of para-cyclists as well as athletes from other para sports are required and we call on sport organisations to collaborate with each other in this area of concussion research to better understand para-athletes’ recovery times following a SRC event.

## 6. Limitations

We have not documented the impact of SRC symptom severity and compared this to length of time to recover from the SRC. We are aware that higher initial symptom severity in the first 72 h post-SRC will cause a more prolonged recovery time from the SRC and would have influenced our results [29]. Additionally, we have not analysed the data for concussion history and the impact that repeated concussions has on return to competition time. Furthermore, two of the concussion cases included within the cohort are still ongoing and we entered an end date of their concussion (the start date of this research study), which was earlier than it should have been. We have therefore under-estimated the duration out of competition in these two cases. Medical staff do not attend all GBCT training sessions and competitions; therefore, some subtle concussions may not have been reported to the BC medical staff. We recognise that one athlete in 2017 was recorded as having a recovery time of 4 days, which is not consistent with GBCT concussion policy. Unfortunately, at the time of data collection, the full details of the concussion episode were not available to be interrogated to clarify the reason for this short recovery period. It is suspected that there may have been a delayed reporting of the injury, with the injury date being recorded incorrectly as the day of presentation rather than the day of injury.

## 7. Conclusions

This is the first study to report SRC concussion recovery times in elite cycling, including para-athletes. Between January 2017 and September 2022, there were 88 concussions diagnosed at BC and the median duration for time out of competition for all concussions was 16 days. There was no statistically significant difference in recovery times between male and females and para and able-bodied athletes. These data should be used to inform minimum withdrawal times post-SRC for elite cycling participation. We call on the UCI to review these data when establishing SRC protocols for cycling and increase the minimum duration out of competition following SRC from the current minimum of 6 days to at least 14 days. Further research is particularly needed in para-cyclists to gain further understanding of their SRC recovery times.

## Figures and Tables

**Figure 1 ijerph-20-05449-f001:**
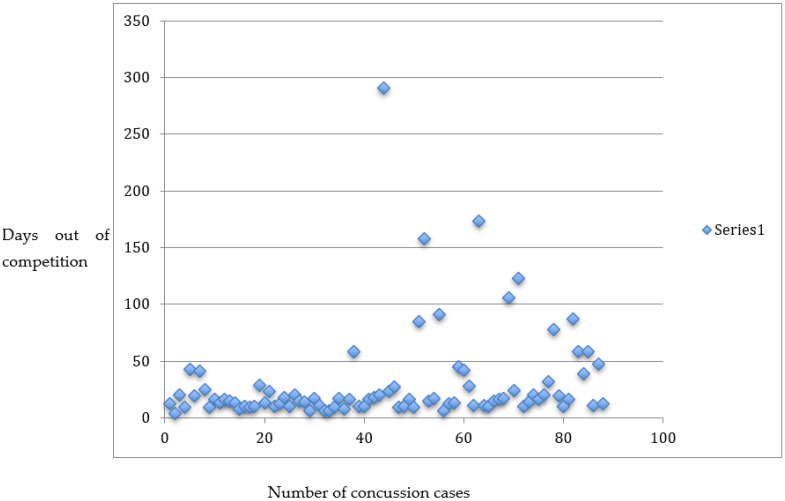
The number of concussions and the days out of competition for each concussion.

## Data Availability

All relevant data has been made available within this publication.

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
