# Peer review of "‘If in Doubt, Sit Them Out’: How Long to Return to Elite Cycling Competition following a Sports-Related Concussion (SRC)?"

_ijerph, 2023, doi:10.3390/ijerph20085449_

Round 1
Reviewer 1 Report
Thank you for your submission. Although there is no statistically significant impact, this paper will be one of the basis for forthcoming studies in this field. Other data with precise factors including past concussion history, concussion severity, comorbidity, etc. should be accumulated and added to this kinds of studies.
Author Response
Thank you for the positive review and comments.
Reviewer 2 Report
This paper provides an interesting communication on SRC in cycling, offering a needed addition to the literature on SRC in cycling. Whilst there is merit in the paper, work is needed for this to be of publishable standard. Please see my comments to be addressed below.
Introduction
I would remove author titles under main title as this is not usual academic convention.
L9: Remove quotation marks and italicisation as a reference is not provided and it reads awkwardly.
L51-54: please provide references.
In the section ‘What is concussion?’ you include some discussion of rates. Whilst this is a can of worms and of course there are many issues with estimate rates of concussion, it would be useful to highlight estimated rates in cycling to contextualise your piece.
In the section ‘Symptoms and risks of concussion’, the authors might consider highlighting Post-concussive syndrome as a subacute risk of SRC. Would seem relevant given some high profile cases in cycling of PCS.
L73-76: This is an excellent point but it would be useful here to build this into more of an academic point with references to literature highlighting these pressures. It would be worth highlighting the pressures to perform and culture of performance over health in elite sporting worlds also.
The section ‘Recommended periods of time out of competition following a concussion’. Please expand or add some discussion on how these arbitrary time periods have been agreed upon. What evidence were these based on? And if the evidence is sloppy as the authors seem to hint at, make this point more explicit.
Methods
Please provide more clarity on the data analysis. Was normality testing run to establish Mann-Whitney U was the correct test? What level was significance set at? What software was used and how was the data managed?
Please provide more clarity on what qualifies as ‘Elite Cyclists’. Also, what disciplines were represented? Would this not be useful to also run analysis on?
Please add if the concussions occurred in racing or training.
Is there a chance concussions were missed? If they occurred in training without medical presence? I would suggest highlighted the possibility of this if so.
Also much more context is required to interpret these results. How many cyclists are on the medical data base? Much more clarity is needed on the affected population. Please provide a table or more information showing the 88 concussions against the affected people. For example, it’s hard to tell how many suffered multiple concussions.
Results
I am not sure if it is a formatting issue with the manuscript downloaded from the journal, but Figure 1 is blank. Please adjust so the data can be clearly seen and reviewed.
Please present further data on rates across disciplines and when they occurred (competition or training). Much more context is required.
Discussion
L166-172: This is a good point. You are referencing to the difference between sex (physiological differences) and gender (how each sex behaves differently in line with societal norms associated with their sex). I would make this point clearer, and perhaps use some of the literature on the influence of masculinity and masculine norms on males under-reporting SRC.
Conclusion
This is mostly repetitive of what’s already been said. I would suggest making this a clearer take home message. So, based on this data, are you suggesting a period of 16 days? If so, state that. Or at least state you wish for the UCI to increase the minimum from 6 days to align with a more precautionary approach to protect athlete health. Basically, this needs more punch and the emphasis the ‘so what’ of this study.
Conflicts of interest.
I see some of the authors are employed by British Cycling. This seems like a conflict of interest to me. Please re-visit this section and adjust accordingly.
Overall comments
Is there not an issue with concussion management and RTP whereby subjective symptoms resolve ahead of neurophysiological damage so athletes may present as ‘fine’ but are not. This problematises RTP and some discussion of this would strengthen the manuscript by being upfront by the limitations in the area.
Author Response
This paper provides an interesting communication on SRC in cycling, offering a needed addition to the literature on SRC in cycling. Whilst there is merit in the paper, work is needed for this to be of publishable standard. Please see my comments to be addressed below.
Thank for the positive review and comments below, which we have addressed in turn.
Introduction
I would remove author titles under main title as this is not usual academic convention.
Author titles have been removed.
L9: Remove quotation marks and italicisation as a reference is not provided and it reads awkwardly.
This has been amended.
L51-54: please provide references.
References provided (5).
In the section ‘What is concussion?’ you include some discussion of rates. Whilst this is a can of worms and of course there are many issues with estimate rates of concussion, it would be useful to highlight estimated rates in cycling to contextualise your piece.
Thanks. I have included:
“Within road cycling, a systematic review reported that head injuries were the 3rd commonest group of injuries, accounting for between 5-15% of all injuries in the sport. Thus concussions are common within the sport of cycling.”
In the section ‘Symptoms and risks of concussion’, the authors might consider highlighting Post-concussive syndrome as a subacute risk of SRC. Would seem relevant given some high profile cases in cycling of PCS.
Thank you. I have now included:
“This includes the risk of developing prolonged concussion symptoms. “
L73-76: This is an excellent point but it would be useful here to build this into more of an academic point with references to literature highlighting these pressures. It would be worth highlighting the pressures to perform and culture of performance over health in elite sporting worlds also.
Thanks. I have added:
“Indeed the psychosocial pressures faced by athletes are increasingly being recognised in affecting athletes’ decisions to return to play, although the physical ‘readiness’ is given priority in current return to play decision making.”
The section ‘Recommended periods of time out of competition following a concussion’. Please expand or add some discussion on how these arbitrary time periods have been agreed upon. What evidence were these based on? And if the evidence is sloppy as the authors seem to hint at, make this point more explicit.
Thank you for this point. I have amended this paragraph to now read:
“Recommended periods of time out of competition following a concussion
Minimum recommended periods of time out of competition in professional sport have been made. These estimates are largely based on expert opinion, starting from the 1st international conference of concussion in sport. When evidence is available it is extrapolated from observing elite male athletes and we know that age affects return to play time, with younger, high school athletes taking longer to recovery from concussions than the older professional athletes. Return to play times differ between sports although the evidence for this is not clear and may be influenced by the timing of the next game following a concussive event. Elite cycling does not have the same pressure to have athletes ready for the following weekend because of the timings of the international competitive cycling calendar, which often has a number of weeks between important competitions. For example, the English Football Association state that the earliest a player aged 20 years or older can return to play is 6 days in an ‘enhanced care setting’ (15) and this approach is supported by the World governing body of football, FIFA (16). This time period of 6 days out of competition post-concussion for adults is similar to what the governing body for cycling recommends, the Union Cycliste International (UCI) (17). However, in professional rugby this period out of competition is now 12 days (18) but what is the evidence for these time periods out of competition? We therefore wanted to address this knowledge gap within elite cycling by reviewing the time period out of competition following a diagnosis of SRC for all elite riders within British Cycling. This is the first study to report recovery times post-SRC within elite cyclists.”
Methods
Please provide more clarity on the data analysis. Was normality testing run to establish Mann-Whitney U was the correct test? What level was significance set at? What software was used and how was the data managed?
Thanks. We have added:
“Data was anonymously plotted in Excel sheets and then analysed in the statistical package, SPSS (IBM SPSS 29). An independent statistician, CC, with input from the lead author, NH, analysed the data. The data was assessed for normality by plotting the results on a histogram. The data was described using medians and minimum and maximum values. Between group differences were tested for using the non-parametric Mann-Whitney U test, with a statistically significant result being p<0.05.”
Please provide more clarity on what qualifies as ‘Elite Cyclists’. Also, what disciplines were represented? Would this not be useful to also run analysis on?
I have added in the Methods section:
“All elite cyclists within the GBCT are full-time athletes, representing their country at the highest level of cycling sport, including national and international events, such as the Olympic and Paralympic Games. There are approximately 120 riders each year included within the GBCT pool of riders, with most undertaking road and track disciplines.”
Please add if the concussions occurred in racing or training.
I have added in the Methods section:
“….and these events could occur in both training and competition.”
Is there a chance concussions were missed? If they occurred in training without medical presence? I would suggest highlighted the possibility of this if so.
This has been added to the Limitations section of the paper.
“Medical staff do not attend all GBCT training sessions and therefore some subtle concussions may have been missed by coaching staff and not reported to the BC medical staff.”
Also much more context is required to interpret these results. How many cyclists are on the medical data base? Much more clarity is needed on the affected population. Please provide a table or more information showing the 88 concussions against the affected people. For example, it’s hard to tell how many suffered multiple concussions.
Thank you. As above we have added to the methods section:
“All elite cyclists within the GBCT are full-time athletes, representing their country at the highest level of cycling sport, including national and international events, such as the Olympic and Paralympic Games. There are approximately 120 riders each year included within the GBCT pool of riders, with most undertaking road and track disciplines.”
As well as to the results section:
“Eighteen riders suffered more than one concussion over the study period, with 1 rider having suffered 4 concussion episodes, 7 riders having 3 concussion episodes and 10 having 2 concussion episodes. Due to the small number of athletes involved, no further analysis was undertaken on riders with multiple concussions.”
Results
I am not sure if it is a formatting issue with the manuscript downloaded from the journal, but Figure 1 is blank. Please adjust so the data can be clearly seen and reviewed.
Figure 1 is included.
Please present further data on rates across disciplines and when they occurred (competition or training). Much more context is required.
We have added that the concussions occur both in training and competition as well as mainly occurring in road and track disciplines.
Discussion
L166-172: This is a good point. You are referencing to the difference between sex (physiological differences) and gender (how each sex behaves differently in line with societal norms associated with their sex). I would make this point clearer, and perhaps use some of the literature on the influence of masculinity and masculine norms on males under-reporting SRC.
We have added:
“Indeed male rugby league players have been found to under report concussive events and symptoms to allow them to continue to play and compete.”
Conclusion
This is mostly repetitive of what’s already been said. I would suggest making this a clearer take home message. So, based on this data, are you suggesting a period of 16 days? If so, state that. Or at least state you wish for the UCI to increase the minimum from 6 days to align with a more precautionary approach to protect athlete health. Basically, this needs more punch and the emphasis the ‘so what’ of this study.
Thank you. I have amended the conclusion to read:
“This is the first study to report SRC concussion recovery times in elite cycling, including para-athletes. Between Jan 2017 and Sept 2022 there was 88 concussions diagnosed at BC and the median duration for time out of competition for all concussions was 16 days. There was no statistically significant difference in recovery times between male and females and para- and able-bodied athletes. This data should be used to help establish minimum withdrawal times post-SRC for elite cycling participation. We call on the UCI to review this data when establishing SRC protocols for cycling and increase the minimum duration out of competition following a concussive event from the current minimum of 6 days to at least 14 days. Further research is particularly needed in para-cyclists in order to gain further understanding of their SRC recovery times.”
Conflicts of interest.
I see some of the authors are employed by British Cycling. This seems like a conflict of interest to me. Please re-visit this section and adjust accordingly.
This now reads:
“Conflicts of Interest: NH, NJ and CG are all employed by British Cycling; there are no other conflicts of interest.”
Overall comments
Is there not an issue with concussion management and RTP whereby subjective symptoms resolve ahead of neurophysiological damage so athletes may present as ‘fine’ but are not. This problematises RTP and some discussion of this would strengthen the manuscript by being upfront by the limitations in the area.
Thanks. We have added to the introduction:
“Another important consideration when determining an appropriate minimum stand-down period is the likelihood that apparent clinical recovery occurs prior to physiological recovery. (19)”
Ref Rooney D, Sarriegui I, Heron N. 'As easy as riding a bike': a systematic review of injuries and illness in road cycling. BMJ Open Sport Exerc Med. 2020 Dec 9;6(1):e000840. doi: 10.1136/bmjsem-2020-000840. PMID: 34422283; PMCID: PMC8323466.
Ref Ryan LM, Warden DL. Post concussion syndrome. Int Rev Psychiatry. 2003 Nov;15(4):310-6. doi: 10.1080/09540260310001606692. PMID: 15276952.
van Ierssel J, Pennock KF, Sampson M, Zemek R, Caron JG. Which psychosocial factors are associated with return to sport following concussion? A systematic review. J Sport Health Sci. 2022 Jul;11(4):438-449. doi: 10.1016/j.jshs.2022.01.001. Epub 2022 Jan 10. PMID: 35017101; PMCID: PMC9338335.
Aubry M, Cantu R, Dvorak J, Graf-Baumann T, Johnston K, Kelly J, Lovell M, McCrory P, Meeuwisse W, Schamasch P; Concussion in Sport Group. Summary and agreement statement of the First International Conference on Concussion in Sport, Vienna 2001. Recommendations for the improvement of safety and health of athletes who may suffer concussive injuries. Br J Sports Med. 2002 Feb;36(1):6-10. doi: 10.1136/bjsm.36.1.6. PMID: 11867482; PMCID: PMC1724447.
D'Lauro C, Johnson BR, McGinty G, Allred CD, Campbell DE, Jackson JC. Reconsidering Return-to-Play Times: A Broader Perspective on Concussion Recovery. Orthop J Sports Med. 2018 Mar 14;6(3):2325967118760854. doi: 10.1177/2325967118760854. PMID: 29568786; PMCID: PMC5858632.
Putukian M, Aubry M, McCrory P. Return to play after sports concussion in elite and non-elite athletes? Br J Sports Med. 2009 May;43 Suppl 1:i28-31. doi: 10.1136/bjsm.2009.058230. PMID: 19433421.
Longworth T, McDonald A, Cunningham C, Khan H, Fitzpatrick J. Do rugby league players under-report concussion symptoms? A cross-sectional study of elite teams based in Australia. BMJ Open Sport Exerc Med. 2021 Jan 19;7(1):e000860. doi: 10.1136/bmjsem-2020-000860. PMID: 33520253; PMCID: PMC7817803.
Reviewer 3 Report
The author could improve the manuscript by adding appropriate references, they make several claims throughout the manuscript that are not cited at all eg line 9, 43, 59.
Figure 1 the data is missing
The manuscript could benefit from significant editing, the authors claim on line #84 they will discuss the evidence for time out periods. This manuscript does not address knowledge gaps within this space. Further, retrospective study designs in this manner are not appropriate to explore this.
Line 112 the author state "there is little evidence for a difference between able-bodied and para-cyclists" despite paracyclis return to competition duration being 3 times longer than able-bodied in their own data. Would a better interpretation not be that they are underpowered to discuss but further research is needed?
Ultimately, this injury surveillance type of research requires very large samples to produce meaningful comparisons. While this data is important, a closer explanation for important variables is required. For example, when did athletes return to training? the return to competition is determined by when the meets are. As discussed in the manuscript for rugby the 7 day period is determined by when the next game is. So if an athlete has not returned by day 7 they will not return for 14 days because there are no other games. In this sample, how many of these cyclists RTP dates were set by competition dates and are not reflective of neurophysiological recovery time?
What was the concussion history or demographics of the sample? how many of the concussed sample were recording their first concussive injuries? If they were subsequent injuries was the recovery period longer as discussed in the manuscript?
Author Response
The author could improve the manuscript by adding appropriate references, they make several claims throughout the manuscript that are not cited at all eg line 9, 43, 59.
Additional references have been added.
Figure 1 the data is missing
Figure 1 is present.
The manuscript could benefit from significant editing, the authors claim on line #84 they will discuss the evidence for time out periods. This manuscript does not address knowledge gaps within this space. Further, retrospective study designs in this manner are not appropriate to explore this.
We feel this paper adds value, supporting more conservative return to play guidelines in cycling and is the 1st study within cycling to detail this, albeit there are potential limitations of the study, that have been recognised.
Line 112 the author state "there is little evidence for a difference between able-bodied and para-cyclists" despite paracyclis return to competition duration being 3 times longer than able-bodied in their own data. Would a better interpretation not be that they are underpowered to discuss but further research is needed?
Thanks. We state that there is no statistical differences between the groups in terms of median return to competition as well as acknowledging the low numbers in the para group (8).
Ultimately, this injury surveillance type of research requires very large samples to produce meaningful comparisons. While this data is important, a closer explanation for important variables is required. For example, when did athletes return to training? the return to competition is determined by when the meets are. As discussed in the manuscript for rugby the 7 day period is determined by when the next game is. So if an athlete has not returned by day 7 they will not return for 14 days because there are no other games. In this sample, how many of these cyclists RTP dates were set by competition dates and are not reflective of neurophysiological recovery time?
We disagree with this comment and hence why this study is actually very informative. As we explain in the introduction, cycling doesn’t have the time pressures of getting athletes ready for the next weekend as events are often spaced throughout the year. Thus these return to play times are very informative and help explain the need for a more conservative approach when managing concussions.
What was the concussion history or demographics of the sample? how many of the concussed sample were recording their first concussive injuries? If they were subsequent injuries was the recovery period longer as discussed in the manuscript?
We report:
“Eighteen riders suffered more than one concussion over the study period, with 1 rider having suffered 4 concussion episodes, 7 riders having 3 concussion episodes and 10 having 2 concussion episodes. Due to the small number of athletes involved, no further analysis was undertaken on riders with multiple concussions.”
And we acknowledge this within our limitations:
“We have not documented the impact of SRC symptom severity and compared this to length of time to recover from the SRC. We are aware that higher initial symptom severity in the first 72 hours post-SRC will cause a more prolonged recovery time from the SRC and would have influenced our results (29).”
Thank you again for the review and your comments.
Yours sincerely,
Dr Neil Heron.
Round 2
Reviewer 2 Report
No further comments.
Author Response
Thank you for the comments and review.
Reviewer 3 Report
I thank the author for their amendments, I have a few minor comments.
Could the authors please make clear the RTP protocol for these athletes? "Medically monitored until full symptom resolution on SCAT and then graduated 6 stage process etc"
Could the authors please confirm when they refer to "return to competition" they mean "return to training" Although cycling competition does not have the 6-day time pressure of rugby as stated in the introduction. This does not preclude their athletes from pressure to return to competition depending on when the injury occurred.
If these athletes were medically instructed through RTP and they do mean "return to training" this manuscript adds a lot to the field.
As stated in the introduction "the more concussions suffered by an individual, the higher the risk of developing these persistent symptoms" Would it be possible to run a regression on the 18 individuals with repeated concussions to explore if repeated concussions are driving the RTP duration?
Author Response
Thank you for the review and further comments.
I thank the author for their amendments, I have a few minor comments.
Could the authors please make clear the RTP protocol for these athletes? "Medically monitored until full symptom resolution on SCAT and then graduated 6 stage process etc"
Thank you for this comment. Within this methods section, we state:
".....with the SRCs diagnosed and managed as per international guidelines (12, 25)."
Could the authors please confirm when they refer to "return to competition" they mean "return to training" Although cycling competition does not have the 6-day time pressure of rugby as stated in the introduction. This does not preclude their athletes from pressure to return to competition depending on when the injury occurred. If these athletes were medically instructed through RTP and they do mean "return to training" this manuscript adds a lot to the field.
Yes it is time to return to full training. I have added to the methods section:
".....full, unrestricted return to training."
As stated in the introduction "the more concussions suffered by an individual, the higher the risk of developing these persistent symptoms" Would it be possible to run a regression on the 18 individuals with repeated concussions to explore if repeated concussions are driving the RTP duration?
We cannot currently do this analysis due to the small number but the plan is to undertake such further analysis within future research projects.